# Selenium-Enriched Spirulina (SeE-SP) Enhance Antioxidant Response, Immunity, and Disease Resistance in Juvenile Asian Seabass, *Lates calcarifer*

**DOI:** 10.3390/antiox11081572

**Published:** 2022-08-14

**Authors:** Muhammad A. B. Siddik, Ioannis N. Vatsos, Md. Arifur Rahman, Hung Duc Pham

**Affiliations:** 1Department of Fisheries Biology and Genetics, Patuakhali Science and Technology University, Patuakhali 8602, Bangladesh; 2Faculty of Biosciences and Aquaculture, Nord University, 8049 Bodø, Norway; 3Faculty Institute of Aquaculture, Nha Trang University, Nha Trang 650000, Khanh Hoa, Vietnam

**Keywords:** microalgae, *Arthrospira platensis*, selenium, histopathology, barramundi, PCA analysis

## Abstract

The present study examined the efficacy of dietary selenium-enriched spirulina (SeE-SP) on growth performance, antioxidant response, liver and intestinal health, immunity and disease resistance of Asian seabass, *Lates calcarifer.* A total of 480 seabass juveniles with an initial weight of 9.22 ± 0.09 g/fish were randomly assigned to four dietary groups. The fish were fed a fishmeal protein replacement diets with SeE-SP at 5%, 10%, and 20%, namely SeE-SP5, SeE-SP10, and SeE-SP20, and a fishmeal-based diet as control for 8 weeks. The results indicated that seabass juveniles fed SeE-SP5 and SeE-SP10 diets grew at the same rate as the fish fed a fishmeal-based control diet after 8 weeks of feeding, while SeE-SP20 grew at a significantly lower rate than the control (*p* < 0.05). Although most of the measured biochemical parameters were not influenced by the Se-SP diets, serum antioxidant-enzyme glutathione peroxidase (GPx) and immunological indices, such as lysozyme activity and immunoglobulin-M, were found significantly higher in the SeE-SP5 and SeE-SP10 diets compared to control. In addition, the fish fed the SeE-SP5 diet showed significantly lower mortalities after the 14-day of bacterial challenge with *V. harveyi*. These outcomes indicated that up to 10% inclusion of SeE-SP in the diet of juvenile Asian seabass does not compromise growth, while SeE-SP5 enhanced disease resistance in juvenile seabass.

## 1. Introduction

Improved and diversified production techniques along, with the intrinsic superior production features of fish, have fueled the aquaculture industry’s rapid rise. As a result, by 2050, fish will be the primary source of food for 9 billion people [1]. However, there are various challenges associated with the production of fish in aquaculture systems, one of which is the vulnerability to disease outbreaks. In this case, the addition of feed additives in the feed formulation of fish, such as probiotic, micronutrients and microalgae, are seen as a promising field in the aquaculture sector [2,3,4].

Selenium (Se) is a key micronutrient that enhances fish health by acting as a potent antioxidant when included in the diet. The antioxidative defenses formed by Se protect cells from oxidation [5]. Furthermore, Se is a component of at least 25 selenoproteins and an important co-factor in the antioxidant enzyme system. Se consumption has an impact on nutrition utilization, productivity, antioxidative mechanisms, reproductive function, hormone metabolism, and immune system responses in productive animals [6,7,8]. In a study by [9], it was found that the yellowtail kingfish (*Seriola lalandi*) fed a diet without Se supplementation showed necrotic myopathy in fish muscles. As a result, including Se nanoparticles into the aquafeed sector is highly recommended for the purpose of enhancing the productivity and health of aquatic animals.

Microalgae have been added to aqua-diets as supplements to promote the growth and health performance of the host fish [10] as a consequence of their high nutritional value [11,12], capabilities for immunomodulation [13], enhancement of pigmentation [14], and as a great source of vitamins and micronutrients [15]. The blue-green microalgae spirulina has a high protein content (up to 70%), a balanced fatty acid profile, vitamins, minerals, and a high concentration of bioactive substances, which helps the aquaculture organisms resist pathogens and environmental stressors [16]. Spirulina has been studied in many aquaculture species and now established as a functional feed ingredient in aquaculture production to improve growth, welfare, and the disease resistance of fish. A study by [17] reported that juvenile seabass performance in terms of growth and health is unaffected when up to 20% of their diets are provisioned with spirulina instead of fishmeal. The growth performance was reported to be augmented at any incorporation level of *S. platensis* in *Labeo rohita* [18]. The addition of *Spirulina* in the diet, comprising of phenolic compounds, improves the antioxidant response of *Oplegnathus fasciatus* [19]. Enzymes, such as glutathione peroxidase (GPx) and catalase (CAT), are important biochemical criteria for the antioxidant defense systems, and dietary selenium and spirulina supplementation have been demonstrated to improve all of these enzymatic activities [20]. *Spirulina* were reported to lower the blood cholesterol levels and promote the production of white and red blood cells in *Oncorhynchus mykiss*, resulting in improved immunity [21].

Asian Seabass is a fast-growing highly carnivore fish and the higher metabolic rates associated with faster-growing fish require sufficient energy to maximize their welfare, resulting in a need to uptake more nutrients, particularly vitamins and minerals. For example, a study with minerals’ supplementation, particularly selenium, with commercial fishmeal improved the welfare of yellowtail kingfish [9]. In addition, the natural immunostimulants e.g., microalgae and carotenoid pigments along with selenium, provide an intriguing alternative approach to immunoprophylactic control, as these compounds play essential roles in modifying and boosting the immunological response of aquatic animals against various pathogens [22]. Therefore, the current study investigated how the dietary supplementation of spirulina, coupled with organic selenium, affects the growth performance, antioxidant response, immunity, and disease resistance of the juvenile Asian seabass.

## 2. Materials and Methods

The current study was performed at Nha Trang University (NTU), Khanh Hoa, Vietnam. The fresh spirulina were collected from the Northwest region of Vietnam and the selenium-enrichment was carried out in the laboratory. The animal care was fully compliant with the Vietnamese Code of Practice for the care of animals for scientific purposes.

### 2.1. Spirulina Enrichment with Selenium

The organic Se (Sel-Plex, Alltech, USA) was added to the spirulina biomass through biosorption. The enrichment procedure was carried out in tanks with 50 L of room-temperature tap water and a selenium solution. The solutions were prepared by dissolving selenium at a rate of 12 mg/L. The contact time was 4 h. The enhanced biomass was then separated, using a filter with a 6 mm pore diameter, dried at 50 °C and crushed. A respective sample of dried feed ingredient of spirulina biomass from each enrichment tank was sampled, stored at −4.0 °C for a week before biochemical analysis.

### 2.2. Dietary Ingredients and Diet Formulation

The feed ingredients, formulation, and their nutrient composition are presented in Table 1. The diets were formulated to contain 0, 5, 10, and 20% spirulina protein named control, SeE-SP5, SeE-SP10, and SeE-SP20 as replacement with fishmeal protein. The diet without selenium supplementation was regarded as the control. Except for the fishmeal, which was supplied by TC Union Vietnam (Tien Giang, Vietnam), all of the feed ingredients were obtained from Long Sinh Feed Company (Khanh Hoa, Vietnam). All of the ingredients were properly mixed together to create a homogeneous mixture. In a food mixer, fish oil and 30% distilled water were added into the premixed dry components and mixed for 15 min to make a dough. The dough was screw-pelletized into 2.0–3.0 mm pellets, using a laboratory pelletizer. These moist pellets were oven-dried for 12 h at 50 °C, and then chilled to an ambient temperature before being kept at −4°C until needed.

### 2.3. Experimental Setup and Facilities

Juvenile seabass was obtained from a commercial marine fish hatchery (Nha Trang, Khanh Hoa, Vietnam) and transported to the wet laboratory in Nha Trang University, where they were acclimated to experimental condition for 2 weeks. During acclimation, the fish were fed Uni-President-Vietnam feed (46% protein and 13% fat). The fish were then graded, and those within the weight range of 9.0–10.0 g were selected and randomly stocked into 12 tanks of 350 L water capacity tank. Each dietary treatment was triplicated, and every tank was stocked with 40 fish. Fish were fed three times a day (8.00, 12.00 and 16.00) until satiation, which took around 20 minutes. Uneaten feed was siphoned off right after feeding to justify feed intake. During the feeding trial, the dissolved oxygen was maintained higher than 5.0 mg/L, average temperature was 29.0 °C, total ammonia was less than 0.25 mg/L and salinity ranged from 29.0 to 32.0 ppt. Following a day of feed deprivation, total numbers of fish in each tank were counted and individual body weights were measured at the end of the growth trial.

### 2.4. Blood and Serum Biochemical Analyses

At the termination of the feeding trial, the blood samples were taken from six fish per tank by puncturing the caudal vein with a 1 mL non-heparinized syringe. For the measurement of the blood hemoglobin, the white blood cells (WBC), red blood cells (RBC), and glucose, an aliquot of the extracted blood samples was collected in heparinized tubes. An automated blood analyzer was used to analyze blood hemoglobin, RBC, and leukocrite (Sysmex XT-162 1800i, Kobe, Japan). The blood was centrifuged in glass capillary tubes at 2000 rpm for five minutes to calculate the blood hematocrit percentage and to test the blood glucose using a blood glucose meter kit (Accu-Chek, Sydney, Australia) [23]. Another set of blood samples was drawn into non-heparinized tubes and left to sit for 24 h before being centrifuged for 10 min at 4 °C at 5000 g to extract the serum. The separated serum samples were maintained at −20 °C until use. The serum blood parameters encompassing alanine aminotransferase (ALT), aspartate aminotransferase (AST), total cholesterol (TC), triglyceride (TG), total protein, and albumin concentrations using an automated blood analyzer (SLIM; SEAC Inc, Florence, Italy), and following the methods of [24]. The serum lysozyme activity was evaluated following the previously described procedure of [25]. An enzyme-linked immunosorbent test was used to assess the immunoglobulin M (IgM), using a commercial kit (Cusabio, Wuhan, Hubei, China). Following the manufacturer’s instructions, the enzymatic activity of the serum glutathione peroxidase (GPx) and catalase (CAT) were measured, using kits (Cusabio Biotech Co., Ltd., Wuhan, China).

### 2.5. Histological Examination

Six fish per treatment (two fish from each replicate) were randomly selected for the histological analysis. The dissected liver and distal intestine were then fixed in 10% buffered formalin for 72 h, after being cleaned with normal saline solution. Following a series of alcohol washes to dry the samples, they were cleaned in xylene and imbedded in paraffin wax for sectioning. The serial sections were cut to the thickness of 5 μm. The sections were stained with hematoxylin and eosin (H&E), covered with a coverslip, and magnified 400 times under a light microscope (Olympus, Germany). Using the onboard camera, the histopathology pictures were obtained (BX40F4, Olympus, Tokyo, Japan). To determine the intestinal micromorphology, including intestinal fold height (IFh), fold width (IFw), goblet cell number (IGCn), and number of adipocyte cells, ten intact intestinal folds from each dietary treatment were studied, as described previously [25,26]. The sections were assessed for anomalies in structure (the relative area of vacuolation in the liver) using the ImageJ software.

### 2.6. Challenge Test with Vibrio Harveyi

Since the growth performance of the fish fed the SeE-SP20 diet was significantly lowered, this group was not considered for the challenge study. The bacterial challenge trial was carried out with SeE-SP5, SeE-SP10, and the control diets, in accordance with the specified protocol of [27]. In brief, 10 fish from each replicate were intraperitoneally injected using a 1-mL syringe fitted with 27-gauge needle with 0.1 mL of pathogenic *V. harveyi* suspension containing LD_50_ = 1.1 × 10^8^ cfu/mL. The blood samples were collected at 24 h and 3 days after the fish were challenged with *V. harveyi*. The fish were considered for cumulative survival counting 14-days after the challenge trial with *V. harveyi*. The clinical symptoms of vibriosis, including a thick layer of mucus on the body’s surface, congested fins, hemorrhages, and ulceration of the skin and muscle tissue were assessed three times daily for 14 days. According to the established technique for fish euthanasia, the fish exhibiting these vibriosis symptoms were euthanized with AQUI-S at a concentration of 175 mg/L for 20 min.

### 2.7. Samples Analyses and Calculations

The growth performance parameters, such as fish weight gain, FCR, total feed intake, and survival were calculated using the following formulas:Weight gain (g/fish)=[final fish weight−initial fish weight]
Specific growth rate (SGR, %⁄d)=[(ln ( final body weight)−ln (pooled initial weight))/Days]×100
Total feed intake (TFI,g/fish/day )=[(dry diet consumed)/(number of fish)]
Feed conversion ratio (FCR)=[(feed given)/(fish weight gain)]
Survival (%)=[(survived fish)/(stocked fish)]×100

### 2.8. Statistical Analysis

The normality and homogeneity of the variances were validated before any statistical analysis. One-way ANOVA was used with post-hoc Turkey’s HSD multiple comparison tests, to establish the differences among the dietary groups. The Kaplan–Meier method was used to construct the survival graph. The statistical significance was calculated at *p* < 0.05. All of the data were presented as mean ± standard error (SE). For the statistical analysis and graph construction, GraphPad PRISM version 8.0 (GraphPad Software, Inc., La Jolla, CA, USA) was used. The principal component analysis (PCA) was used separately on the datasets obtained from the four dietary groups and various parameters analyzed (growth, blood, antioxidant and histomorphology) at the end of feeding trial, to assess the overall covariation of their respective variables. The PCA analysis was performed in the open-source environment R version 3.6.2 (R Core Team, Vienna, Austria).

## 3. Results

### 3.1. Growth Performance and Feed Utilization

The growth performance and feed utilization of the juvenile seabass fed the SeE-SP diets at varied inclusion levels were significantly affected, except for survival (Table 2). The fish fed SeE-SP5 and SeE-SP10 diets revealed equivalent growth performance in terms of final body weight (FBW) and weight gain (WG) when compared to the control (Figure 1). Whereas, the SeE-SP20 diet produced significantly lower growth performance compared to the control and the other dietary groups. Although the dietary groups of SeE-SP5 and SeE-SP10 produced a significantly higher specific growth rate (SGR) compare to Se-SP20, they were equivalent to the control (*p* > 0.05). The SeE-SP5 diet produced a significantly higher feed intake compared to the control and SeE-SP20 (*p* < 0.05). The SeE-SP20 diet had a significantly higher FCR than the other experimental diets (*p* < 0.05). The survival rate of fish was not impacted by any of the dietary groups (*p* > 0.05). The quadratic regression analysis revealed the highest SeE-SP level of 7.40% for the maximum growth performance of juvenile Asian seabass.

### 3.2. Blood and Serum Biochemical Responses

The serum biochemical parameters including hematocrit, RBC, hemoglobin, glucose, cholesterol, albumin, globulin, total protein, and triglyceride were not significantly (*p* > 0.05) influenced by the SeE-SP diets when compared to the control (Table 2). The enzymatic GPx activity was substantially (*p* < 0.05) increased with the SeE-SP5 and SeE-SP10 diets when compared to the control, while serum CAT activity was lowered in the fish fed the SeE-SP5 diet compared to the control and the rest of the diets (Figure 2). In terms of hepatic enzymatic activities, the serum AST level was found lowered in the fish fed the SeE-SP5 diet, whereas the ALT level was lowered both in the SeE-SP5 and SeE-SP10 diets when compared to the control (Figure 2).

### 3.3. Histological Examination

The liver, intestine, and fat cells micromorphology of the juvenile seabass fed the SeE-SP diets at various inclusion levels are shown in Figure 3. No aberrant hepatocytes were noticed in the livers of any of the dietary groups, except for the SeE-SP20 diet. The fish fed the SeE-SP20 diet produced higher lipid vacuolization when compared to the other dietary groups (*p* < 0.05). The IFh was not impacted up to 10% of the fishmeal replacement diets with SeE-SP diets, but was lowered at the 20% replacement level. Meanwhile, the IFw was neither increased or decreased by the inclusion of the selenium-enriched spirulina diets in the seabass. The IGCn was found to be significantly higher in the SeE-SP5 and SeE-SP10 diets when compared to the control. The fish fed the SeE-SP5 and SeE-SP10 diets were found to be have smaller and higher number of adipocyte cells when compared to the fish fed the control (*p* < 0.05).

### 3.4. Disease Resistance of Fish

The disease resistance of the fish fed the control and the SeE-SP diets was assessed using a pathogen challenge with *V. harveyi*. After 14 days of IP challenges, the fish fed the Se-SP5 diet had considerably lower mortalities than the fish fed the control and other Se-SP10 diets (Figure 4). The mortalities began from day 2 post-challenge, with a mean cumulative mortality of 78.25% in the control, 58.35% in the Se-SP10 and 24.55% in the Se-SP5, 14 days post-challenge.

### 3.5. Immunity of Fish against V. harveyi

The immunological indices including serum lysozyme activity, total protein and IgM of fish were affected both by the dietary groups and the blood sampling periods after challenge with *V. harveyi* (Figure 4). The fish in each dietary group had higher lysozyme activity at 24 h post-challenge in comparison to the pre-challenge condition and the post-challenge condition at 72 h. The IgM level was found to be significantly higher in the SeE-SP5 group, while total protein was higher in both the SeE-SP5 and SeE-SP10 groups compared to the control. The total protein level was unaffected by the challenge, however the IgM levels were greater at 24 h post-challenge compared to the pre-challenge and 72 h post-challenge conditions.

### 3.6. PCA Analysis

The principal component analysis (PCA) was performed to investigate the effects of the dietary groups on various growth, immunity, and health parameters in fish. The first two principal component axes (PC1 and PC2) explained more than 65.3% of the variation in the data, providing information about the predominant correlations between these data. The fish fed the control and the SeE-SP20 diets were grouped together in the PCA biplot’s negative site, where the FCR was negatively correlated with AST, and glucose (Figure 5a). On the other hand, the SeE-SP5 and SeE-SP10 diets grouped together in the PC1 positive zone, where FBW and SGR were found to be highly positively correlated with IFh, IGCn, ACn, TP, IgM, GPx, and lysozyme activity (Figure 5b). This arrangement of the variables demonstrates that the first principal component, which accounts for 52.6% of the variation in all of the parameters, reflects a composite perspective of the majority of the parameters. Additionally, the first principal component (PC1) predicts 52.6% variance, second principal component (PC2) 12.7%, and the third principal component (PC2) 10.52%, and so on. The bi-plot showed that the SeE-SP5 and SeE-SP10 diets were intermingled and overlapped with positive correlation, while the control and SeE-SP20 diets formed distinct groups that were negatively associated with each other.

## 4. Discussion

Spirulina has been put into practice as a fishmeal replacement ingredient in aquafeed due to it higher content of amino acids and micronutrients, such as potassium, magnesium, calcium, zinc, phosphorous, and iron in the dried biomass [28,29]. It also contains immunostimulatory and antiviral properties, ensuring the potential of increasing disease resistance in fish [30]. Meanwhile, selenium is an essential co-factor in the antioxidant enzyme system, whose intake has an impact on nutrition utilization, production performance, antioxidative mechanisms, reproductive function, and immune system responses in fish [31,32]. The past studies have indicated that the demand and bioavailability of dietary selenium is species-specific and its level is regulated by a range of factors, including type of feed, culture condition, age of species to be cultured and the type of selenium itself—whether it is organic or synthetic [33,34]. High mortalities, histological alterations in liver tissues, decreased reproductive performance, and reduction in feed intake, growth, and hematocrit levels are all indicators of an insufficiency of selenium in fish. On the other hand, excessive inclusion levels may result in selenium toxicity, which can damage fish growth and health [33]. A study by [35] found that higher doses of selenium cause an excessive build-up of Se in the liver and kidneys, which may result in oxidative stress in fish [35]. According to the present findings, the selenium-enriched spirulina had a positive influence on growth performance and feed utilization up to a 10% replacement level, however feeding at an exclusive level (>10%) inhibited the juvenile seabass growth. The spirulina supplementation in the range of 1 to 10% has been reported as a viable option for increasing the nutritional value of aquafeed [36], which validates our findings. The parrotfish, *Oplegnathus fasciatus*, treated with 5% Spirulina in formulated feed showed considerably higher weight increase, protein efficiency ratios, feed intake, and lower feed conversion ratios when compared to the fishmeal-fed control [19]. Similarly, rainbow trout, *Oncorhynchus mykiss* gained the most weight when the fishmeal was replaced with 7.5% spirulina [37]. An addition of up to 20% of the fishmeal replaced with spirulina was reported to be without adverse effect on the growth of the golden barb, *Puntius gelius* [38]. Conversely, the growth rate of the silver seabream, *Rhabdosargus sarba*, remained unaltered when 50% of the fishmeal was substituted by *Spirulina maxima* [39], whereas Atlantic cod, *Gadus morhua*, exhibited poor growth when the combination of dried *Nannochloropsis* sp. and *Isochrysis* sp. algae were added to the diet at a 30% inclusion level [29]. The weight gain and feed conversion ratio of juvenile barramundi were not negatively affected by an up to 20% fishmeal replacement with raw spirulina, while feeding at an exclusive level of 40% significantly reduced the growth performance [28]. The mode of action elucidating the variations in weight loss due to diets is unknown, but Refs. [11,40] reported that better lipid mobilization to produce energy delivered by microalgae in the muscle that is accessible from the algae could be the reason for the reduction in weight loss. The Atlantic cod, (*Gadus morhua*) a carnivorous species, when fed spirulina as a fishmeal substitute diet, had a decreased capacity to operate at optimum levels due to the diminished palatability of the meals [29].

The antioxidant defense system, which is maintained through enzymes and the antioxidant state, is closely linked to fish physiological functions and immunity [41]. In fish, there is a cellular equilibrium between the synthesis and clearance of reactive oxygen species (ROS) under normal conditions. When the fish are stressed, high quantities of reactive oxygen species (ROS) are created, which can cause considerable damage to the cell structures [42]. Under this circumstance, the fish activate an antioxidant response to prevent excessive ROS, which reduces the negative effects on the cells and tissues [41]. The antioxidant GPx activity of the blood serum was significantly increased in the selenium-enriched spirulina groups as compared to the control group, according to the findings of this study. This could be because the symbiotic-fed fish have a synchronized effect of selenium and spirulina supplementation, which leads to greater cellular competence against oxidative stress, a decrease in lipid peroxidation, and a higher number of healthy cells in an increasing GPx function. The selenium and spirulina include antioxidants that can suppress the generation of free radicals, boost the uptake mechanism of endogenous radicals, and promote cellular antioxidant enzymes, such as CAT and GPx, according to a number of studies [43,44].

The blood indices are used to determine metabolic activity, nutritional health, and the physiological state of the fish [25,32,45]. A higher quantity of serum protein, albumin, or globulin is associated with a stronger innate immune response, and is recognized as a reliable predictor of immune system activation [45,46]. Although these blood parameters were not significantly improved compared to the control group in this study, their apparent rise indicates that the selenium-enriched spirulina has a good impact on the health of the fish. The blood parameters of barramundi (*Lates calcarifer*) [47] were reported to be influenced by dietary canola meal, while an optimum growth performance was achieved as reported in barramundi [48] or grouper [49] which is in agreement with the present study. On the other hand, adding 10% *S. platensis* into rainbow trout, *O. mykiss*, meals considerably raised the levels of hemoglobin, total protein, albumin, white blood cells, and red blood cells, making it an immunostimulant [21]. The results of the present study revealed that, even at higher replacement levels (>10%), these blood and serum indices were not decreased in the Asian seabass fed selenium-enriched spirulina which confirm the role of selenium and spirulina in balancing the body homeostasis in this species. The overall increase in the serum lysozyme activity in the fish fed the SeE-SP5 diet suggests a stronger immunological response; because the antimicrobial peptides, such as lysozyme, were shown to aid in the inhibition of microorganism colonization in the host body, resulting in pathogen prevention and immune cells to fight infection [50,51]. Clinically, the serum AST and ALT levels are routinely used as indicators for liver health. The increase in the AST and ALT levels indicate hepatic cell damage or increased liver enzyme production [52]. In the current study, the AST and ALT levels were lowered in the fish with the SeE-SP5 diet indicates that the minimum inclusion of spirulina improves the liver condition of fish. It is reported in poultry that the dietary supplementation with the organic selenium improved the antioxidant status and decreased the level of these enzymes’ secretion in the laying hens [53]. The present results are in line with [54], who found the lowest AST and ALP levels in the fish supplemented with selenium and methionine.

The histological assay of the hepatic micromorphology of juvenile Asian seabass revealed abnormal hepatocytes in the dietary group of SeE-SP20, characterized by the formation of lipid droplets. In contrast, a study by [28] found that in the fish given up to 20% replacement of fishmeal *Spirulina platensis* no histological changes were induced in the liver of *L. calcarifer*, whereas a 40% replacement resulted in a higher number of vacuoles and necrotic cells in the liver. The fat accumulation in the hepatocytes occurs when the dietary lipids exceed the ability of the hepatic cells to break them down [55,56]. The available literature indicates that fat accumulation in the liver impedes the fish development and immunological response [55,56]. Regarding the intestine, the fish fed selenium-enriched spirulina diets up to 10% had a significantly higher number of goblet cells and enhanced fold heights, indicating a higher nutrient absorption and utilization in the fish at a moderate inclusion level. The intestinal fold heights were reduced in the SP20 diet, indicating that an overabundance of nutrients disrupted the intestinal homeostasis, resulting in reduced nutritional absorption and, as a result, reduced fish-growth performance. In a study on rainbow trout, it is reported that the goblet cell density, villus height, absorption surface area, and intraepithelial lymphocytes were modulated when the fish were fed a 5% spirulina-supplemented diet [30]. However, a higher inclusion level could lead to the loss of some amino acids, such as lysine, methionine, histidine, arginine, and threonine [57,58,59], which could affect the internal tissues integrity. In addition, the inadequate feeding and subsequent absorption in fish can result in changes in the intestinal structure, such as a decrease in the IGCn numbers and a shortening of the intestinal fold height, which can lead to impaired immune function [25]. On the other hand, increased fold height in the intestine may be due to improved nutrient absorption and utilization [25], which is in fair agreement with the present findings. The current study also reported that the fish given the SeE-SP5 and SeE-SP diets had a smaller but higher number of adipocyte cells, so that the number of adipocyte cells in the SeE-SP5 and SeE-SP10 groups were significantly higher than the control. There are no previous studies to compare with the current study on the effect of microalgae supplementation on adipocyte cell size and number in fish. More research is needed to confirm the effects of microalgae, i.e., spirulina on intraperitoneal fat cells and/or lipid metabolism in fish.

The spirulina extracts, namely, ethanol, acetone, diethyl ether, and methanol, have shown antibiotic efficacy against *Escherichia coli*, *Staphylococcus aureus*, and *Pseudomonas aeruginosa* and are reported to be efficient against *Staphylococcus aureus*, *E. coli*, *Candida albicans*, and *Aspergillus niger* [60,61]. In the present study, after 14 days of IP challenges, the fish given the SeE-SP5 and SeE-SP10 diets showed lower mortalities than the fish fed the control diet when challenged with *V. harveyi*. This suggests that supplementing fishmeal diets with spirulina and selenium could protect fish against bacterial infection. This research discovered a link between dietary SeE-SP and lysozyme activity, which led to increased disease resistance against infections. The therapeutic potential of garlic as a dietary supplement is reported to reduce the mortality of *L. calcarifer* after being challenged with *V. harveyi* [62]. Similarly, a number of studies utilizing spirulina and/or selenium have beem shown to enhance disease resistance against pathogenic bacteria in fish [63,64,65]. It has been reported that spirulina inclusion at an optimum level limits the growth of opportunistic pathogens by giving critical nutrients to the fish [17]. It is also worth highlighting that selenium plays a role in cellular antioxidant activities by increasing the synthesis of selenium-containing enzymes, such as GPx. In fish, dietary selenium can boost GPx activity, and play a key role in eliminating hydrogen peroxide (H_2_O_2_) from the host body to combat stress or infections [66]. An increased GPx scavenging capacity to eliminate hydrogen peroxide diffused from phagolysosomes could be one way by which the dietary selenium improves lysozyme activity in fish [67]. Increased GPx activity during phagocytosis may lower H_2_O_2_ generation and boost the circulation of NADPH oxidases [68].

The PCA analysis was applied to investigate the effect of SeE-SP diets on various growth, immunity, and health parameters in fish. The position of the dietary groups and variables in the plot showed that the SeE-SP5 and SeE-SP10 diets had a favorable influence on fish growth and wellbeing, whereas the SeE-SP20 diet had a negative association with fish performance. The PCA analysis was applied to investigate the effect of the SeE-SP dietary groups on various growth, immunity, and health parameters in fish. Previously, the PCA was also used to assess the data from the whole range of digestive enzyme activity in various species of annelida, in order to determine their dietary preferences [69]. In the present experiment, the position of the dietary groups and growth, immunity, and health parameters in the plot showed that the SeE-SP5 and SeE-SP10 diets had a favorable influence on fish growth and overlapped each other, whereas the SeE-SP20 diet had a negative association with fish performance. Indeed, the SeE-SP5 and SeE-SP10 diets significantly influenced IFh and IGCn, indicating that these diets enhanced intestinal health. The PCA revealed that the fish meal replacement with soy protein concentrate at a level of 50% did not affect the growth of the juvenile lumpfish (*Cyclopterus lumpus*) and up to 70% did not affect the IFH and IGCn [70]. However, in our study, the SeE-SP supplementation at rates of SeE-SP5 and SeE-SP10 was observed to positively connect to IFH and IGCn, while the control and SeE-SP20 negatively impacted these variables.

## 5. Conclusions

The optimum selenium-enriched spirulina (SeE-SP) level for juvenile Asian seabass was estimated to be 7.4%, based on the quadratic regression analysis of the final body weight of fish. Although the growth performance was not significantly enhanced by the SeE-SP inclusion in the diets, the increased antioxidant response, intestinal micro-morphological parameter, and immunological indices, such as lysozyme activity and immunoglobulin-M, were found to be significantly higher in the fish fed up to 10% SeE-SP in diet. Moreover, the fish fed a 5% SeE-SP diet showed significantly lower mortalities after 14-days of bacterial challenge with *Vibrio harveyi*. These outcomes indicate that up to 10% inclusion of SeE-SP in the practical diet does not compromise growth, whilst SeE-SP levels at 5% enhanced the disease resistance in Asian seabass.

## Figures and Tables

**Figure 1 antioxidants-11-01572-f001:**
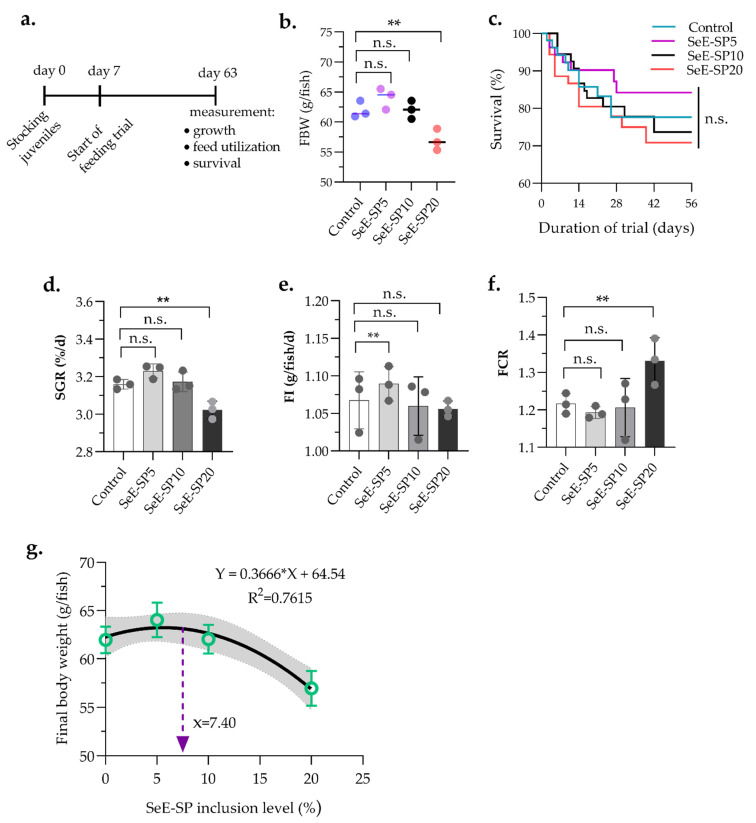
Growth performance and feed intake of juvenile seabass fed various levels of selenium-enriched spirulina (SeE-SP). (**a**) Timeline of feeding trial followed by seven days of conditioning fish in feeding tank; (**b**) Final body weight (FBW); (**c**) Survival; (**d**) specific growth rate (SGR); (**e**) feed intake (FI); (**f**) feed conversion ratio (FCR) of fish after 56 days of feeding trial; (**g**) the highest SeE-SP level for the maximum FBW was 7.40% in the diet, as determined by quadratic regression analysis. Values are mean ± SE of three replicate tanks (indicated as dots on bars) per treatment. The bars with asterisks (**) are significantly different based on Tukey’s multiple range test (One-way ANOVA, *p* < 0.05). ns, non-significant at *p* < 0.05.

**Figure 2 antioxidants-11-01572-f002:**
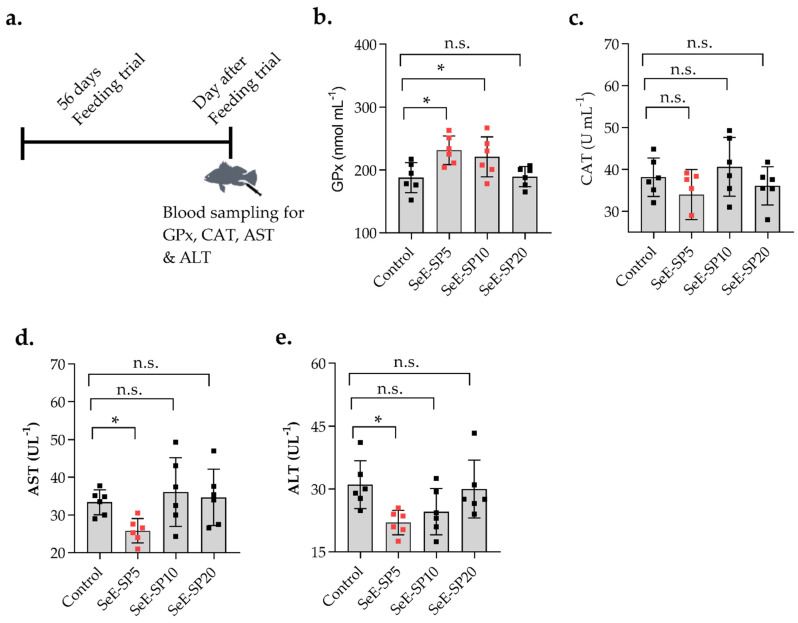
Serum antioxidant response and liver enzymatic activities of juvenile seabass fed various levels of selenium-enriched spirulina (SeE-SP). (**a**) Timeline of feeding trial and blood collection; (**b**) glutathione peroxidase (GPx); (**c**) catalase (CAT); (**d**) aspartate aminotransferase (AST); and (**e**) alanine aminotransferase (ALT) of fish following 56 days of experimental feeding. Values are mean ± SE of three replicate tanks per treatment (values of individual fish per group are indicated as small squares on the bars). Bars holding asterisk (*) are significantly different based on Tukey’s multiple range test (One-way ANOVA, *p* < 0.05). ns, non-significant at *p* < 0.05.

**Figure 3 antioxidants-11-01572-f003:**
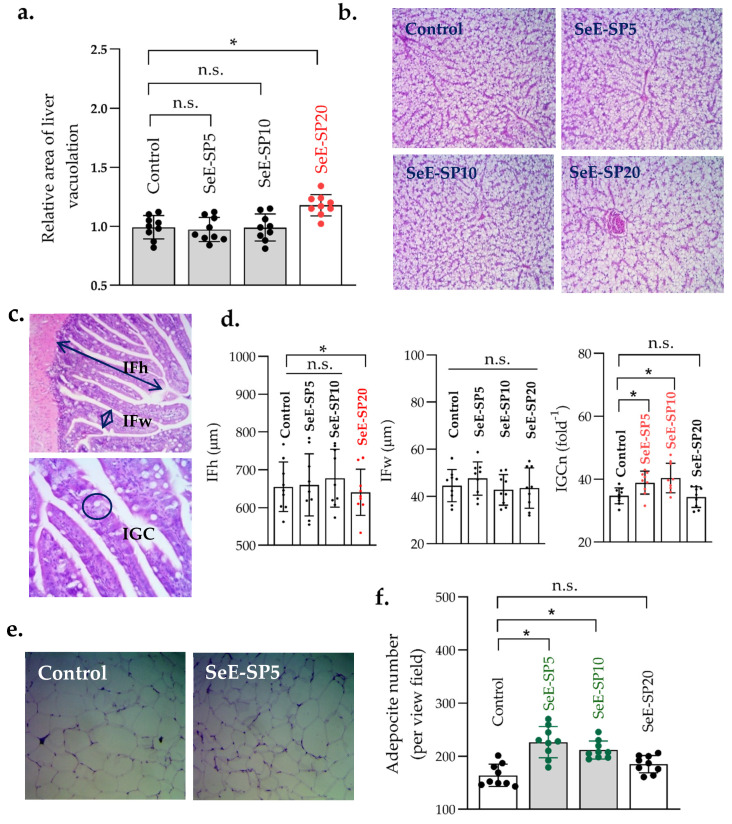
Liver and fat cells histomorphology of juvenile seabass fed various levels of selenium-enriched spirulina (Se-SP). (**a**) Measurement of liver vacuolation; (**b**) representative hepatic images of fish fed various levels of SeE-SP diets (H&E, 40 × magnification); (**c**) Schematic representation of histometric measurement of intestinal fold height (IFh), fold width (IFw) and intestinal goblet cell number (IGCn) (H&E, 40 × magnification); (**d**) quantification of IFh, IFw, and IGCn of the distal intestine of seabass (values are mean ± SE of three replicate tanks per treatment, while values of individual fish per group are indicated as dots on the bars); (**e**) representative adipose tissue (fat cells) histology (H&E, 40 × magnification); and (**f**) quantification of adipocyte number in fish fed various levels of SeE-SP diets (values are mean ± SE of three replicate tanks per treatment, while values of individual fish per group are indicated as small dots on the bars). Bars holding asterisk (*) are significantly different based on Tukey’s multiple range test (One-way ANOVA, *p* < 0.05). ns, non-significant at *p* < 0.05.

**Figure 4 antioxidants-11-01572-f004:**
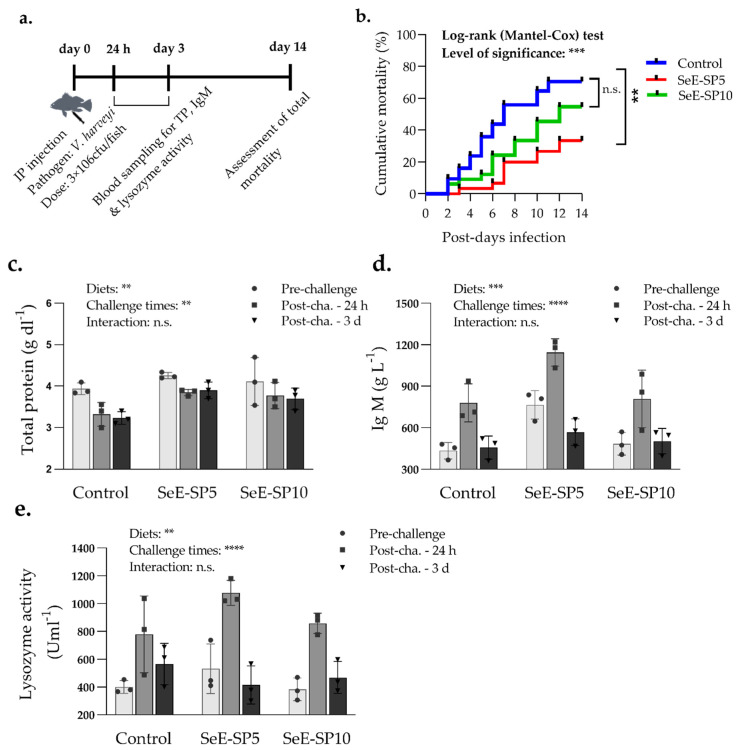
Serum immunological response and mortalities of seabass juveniles challenged with the *Vibrio harvei* after 56 days of the feeding trial (**a**) Timeline of blood collection and mortality assessment followed by *V. harveyi* infection; (**b**) Survival curve of juvenile seabass after challenge with *V. harveyi* for a period of two weeks (Kaplan–Meier survival method, followed by Log-rank test, *p* < 0.05); (**c**) total protein; (**d**) immunoglobulin M (IgM); and (**e**) lysozyme activity of fish at pre-challenge and post-challenge of fish at 24 h and 3 day. Bars holding various asterisks (** ), (***) and (****) indicates whether any significant variation amongst diets, time of sampling and interaction between diets and sampling times (two-way ANOVA; Tukey post-hoc test) at *p* < 0.05, *p* < 0.01 and *p* < 0.001 respectively. n.s., indicates non-significant differences at *p* < 0.05.

**Figure 5 antioxidants-11-01572-f005:**
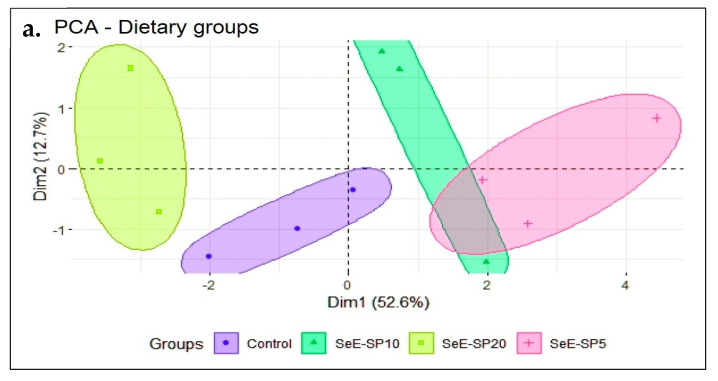
PCA score and biplot with various dietary groups and some measured parameters at the end of the feeding trial. The graph shows positive and negative association between various dietary groups (**a**) and variables (**b**). Different markers with confidence eclipse indicate samples from various dietary groups utilized in the investigation, and loadings show how the variable contributed to the development of PC1 and PC2. FBW, final body weight; SGR, specific growth rate; FCR, feed conversion ratio; IFh, intestinal fold height; IGCn, intestinal goblet cell numbers; TP, total protein; IgM, immunoglobulin M; AST, aspartate aminotransferase, GPx, glutathione peroxidase; and ACn, adipocyte cell number.

**Table 1 antioxidants-11-01572-t001:** Formulation and proximate composition of the experimental diets for juvenile seabass.

Ingredients (g kg^−1^)	Experimental Diets			
	Control	SeE-SP5	SeE-SP10	SeE-SP20
Fishmeal	460.0	438.0	418.0	375.0
Selenium-enriched Spirulina	-	21.0	42.0	84.0
Wheat gluten	120.0	120.0	120.0	120.0
Wheat flour	200.0	245.0	201.0	182.0
Wheat starch	30.0	30.0	30.0	30.0
Fish oil	50.0	50.0	50.0	50.0
Calcium carbonate	2.0	2.0	2.0	2.0
Sodium chlorite	2.0	2.0	2.0	2.0
Vitamin premix	1.0	1.0	1.0	1.0
Cellulose	135.0	81.0	134.0	154.0
Proximate composition				
Dry matter	89.95	91.58	90.90	91.22
Crude protein	45.30	44.98	45.12	45.28
Crude lipid	10.06	10.22	10.24	10.13
Ash	11.19	9.54	9.26	10.70
Gross energy (MJ kg^−1^)	19.90	19.89	19.94	19.91

**Table 2 antioxidants-11-01572-t002:** Blood and serum biochemical parameters and enzymatic activities of juvenile seabass fed various levels of SeE-SP diets for 8 weeks.

Parameters		Experimental Diets			*p*-Value
	Control	SeE-SP5	SeE-SP10	SeE-SP20	
RBC (×10^12^/L)	2.01 ± 0.37	1.4 ± 0.32	1.99 ± 0.39	2.03 ± 0.31	0.425
WBC (10^3^ µL^−1^)	7.89 ± 1.24	7.17 ± 1.01	8.37 ± 1.20	7.11 ± 0.86	0.523
Hematocrit (%)	38.02 ± 3.92	41.64 ± 3.66	40.48 ± 7.32	39.67 ± 7.18	0.834
Hemoglobin (g/dl)	68.66 ± 3.51	74.0 ± 3.75	72.0 ± 3.0	69.67 ± 2.52	0.626
Glucose (nmol/L)	5.45 ± 0.11	5.05 ± 0.51	5.19 ± 0.44	5.23 ± 0.96	0.877
Total protein (g/L)	35.34 ± 0.57	39.87 ± 0.93	37.46 ± 0.93	36.82 ± 4.82	0.761
Albumin (g/L)	12.66 ± 2.52	12.76 ± 2.04	11.42 ± 2.51	13.64 ± 4.72	0.163
Globulin	7.57 ± 5.27	10.20 ± 4.00	9.29 ± 4.29	10.14 ± 5.13	0.742
Cholesterol (nmol/L)	1.30 ± 0.10	1.45 ± 0.42	1.56 ± 0.36	1.68 ± 0.33	0.624
Triglyceride (nmol/L)	1.27 ± 1.41	0.73 ± 0.54	0.76 ± 0.81	0.08 ± 1.03	0.067

Note: Results are presented as the mean (*n* = 6) with standard error. Means without any superscript letters in the same row are not significantly different at *p* > 0.05. RBC, red blood cell; WBC, white blood cell.

## Data Availability

The data presented in this study are available in the article.

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
