# Peer review of "Selenium-Enriched Spirulina (SeE-SP) Enhance Antioxidant Response, Immunity, and Disease Resistance in Juvenile Asian Seabass, Lates calcarifer"

_antioxidants, 2022, doi:10.3390/antiox11081572_

Round 1
Reviewer 1 Report
The work is interesting. Relevant results have been obtained with a perspective for implementation in practice. I have only small wishes to the authors.
1. Figure 1b numbers to the axes need to be corrected
2. PCA analysis and Figure 5 should be described in more detail in the text.
3. The conclusion should be written in more detail and reflect the results obtained in the study
4. The toxic effects of selenium should be discussed. As well as the effects of insufficient selenium intake by the population should be discussed. https://pubmed.ncbi.nlm.nih.gov/34639150/ https://pubmed.ncbi.nlm.nih.gov/34205571/
Author Response
Comments and Suggestions for Authors
The work is interesting. Relevant results have been obtained with a perspective for implementation in practice. I have only small wishes to the authors.
Response: Thank you for the positive remarks on our study
- Figure 1b numbers to the axes need to be corrected
Response: Revised
- PCA analysis and Figure 5 should be described in more detail in the text.
Response: The result and discussion sections related to PCA analysis is thoroughly discussed. Please see the marked text about PCA.
- The conclusion should be written in more detail and reflect the results obtained in the study
Response: The conclusion part is re-written more elaborately based on the obtained result.
- The toxic effects of selenium should be discussed. As well as the effects of insufficient selenium intake by the population should be discussed. https://pubmed.ncbi.nlm.nih.gov/34639150/ https://pubmed.ncbi.nlm.nih.gov/34205571/
Response: the suggested information is included in the discussion section and marked as red color (please see lines 449-458)
Reviewer 2 Report
Auhtors presented a convincng study that addition of Se can influence some of the metabolic oxidative pathways in juvenile asean seabass. Its role in disease resistance is not elucidated and discussion leaves the reader to be less than convinced of the significance of the findings for the bropader audience. In the area of aquaculture, the results are positive and convincingly improving the health/performace of the fish at lower ranges of selenium suppolementation through spiurulina. What is missing from the aquacutlure stanbdpoint is economical viablity of such supplementaiton to fish feed, as there are plenty of other dietary supplements/additives with similar mild improvements in performance.
Line 89 - 6 mm, not 6 m pore diameter.
Lines 91, 103 - was it minus 4 C or 4 C storage? if -4, how did you achieve it? Mostr freezers are set for -18 C, -30 C or lower temp? If 4 C (and if -4 C), how long did you keep the sample (and feed pellets) before analyzing/feeding? Is -4 C sufficient low temp to prevent fat/oil degradation/spoilage within this time?
In table 1 - Are you sure that Sodium chlorite (NaClO2) was used in fish feed at 2g/Kg - Did you mean Sodium Chloride (table salt - NaCl)?
LIne 109 then 117 in LIne 109 you mention acclimation to 5 ppt salinity. In line 117, you say fish were kept at 29-32 ppt salinity. There was no mention in between these lines that further increase in salinity was performed? What is correct? Further explanation may be needed to clarify fish rearing conditions.
Graphical/figure quality can be improved, e.g. formatting for emphasis on differences may be needed as there are multiple parameters that were not significantly different. Therefore, reduction of non-significant clutter may be achieved by compiling the significantly different parameters on a single figure (or single figure with several plots - a,. b, c).
Overall, the manuscript is solid, however the relevance for antioxidants audience is not high.
Acceptance after revisions.
Author Response
Authors presented a convincing study that addition of Se can influence some of the metabolic oxidative pathways in juvenile Asian seabass. Its role in disease resistance is not elucidated and discussion leaves the reader to be less than convinced of the significance of the findings for the broader audience. In the area of aquaculture, the results are positive and convincingly improving the health/performance of the fish at lower ranges of selenium supplementation through spiurulina. What is missing from the aquaculture standpoint is economical viability of such supplementation to fish feed, as there are plenty of other dietary supplements/additives with similar mild improvements in performance.
Response: Thank you for your insightful remarks on the study.
Line 89 - 6 mm, not 6 m pore diameter.
Response: It would be 6 mm and corrected.
Lines 91, 103 - was it minus 4 C or 4 C storage? if -4, how did you achieve it? Most freezers are set for -18 C, -30 C or lower temp? If 4 C (and if -4 C), how long did you keep the sample (and feed pellets) before analyzing/feeding? Is -4 C sufficient low temp to prevent fat/oil degradation/spoilage within this time?
Response: The dried powder of spirulina biomass was stored at -40C for a week or less before bringing to analysis following published literature.
In table 1 - Are you sure that Sodium chlorite (NaClO2) was used in fish feed at 2g/Kg - Did you mean Sodium Chloride (table salt - NaCl)?
Response: It is table salt – NaCl.
Line 109 then 117 in Line 109 you mention acclimation to 5 ppt salinity. In line 117, you say fish were kept at 29-32 ppt salinity. There was no mention in between these lines that further increase in salinity was performed? What is correct? Further explanation may be needed to clarify fish rearing conditions.
Response: The salinity was maintained around 5 ppt throughout the experimental period. It is corrected in the manuscript (same lines).
Graphical/figure quality can be improved, e.g. formatting for emphasis on differences may be needed as there are multiple parameters that were not significantly different. Therefore, reduction of non-significant clutter may be achieved by compiling the significantly different parameters on a single figure (or single figure with several plots - a,. b, c).
Response: All graphs and figures are improved a bit for better visualization. We also added a graphical abstract in the manuscript. However, reduction of non-significant clutter from the graphs would impact overall presentation of data, therefore, we would request you to consider present version that have been revised a bit.
Overall, the manuscript is solid, however the relevance for antioxidants audience is not high.
Response: We have used selenium enriched-spirulina as an antioxidants source in fish feed and result proved with GPx marker that selenium enrichment improved antioxidant status in fish that might influence to improve disease resistance in fish. Therefore, we believe it is very relevant.